# Effects of High-Intensity Motor Learning and Dietary Supplementation on Motor Skill-Related Physical Fitness in Thin Ethiopian Children Aged 5 to 7 Years: An Exploratory Pilot Cluster-Randomized Trial

**DOI:** 10.3390/nu17010030

**Published:** 2024-12-25

**Authors:** Melese Sinaga Teshome, Eugene Rameckers, Sarah Mingels, Marita Granitzer, Teklu Gemechu Abessa, Liesbeth Bruckers, Tefera Belachew, Evi Verbecque

**Affiliations:** 1Department of Nutrition and Dietetics, Faculty of Public Health, Health Institute, Jimma University, Jimma 378, Ethiopia; teferabelachew2@gmail.com; 2Rehabilitation Research Centre (REVAL), Rehabilitation Sciences and Physiotherapy, Hasselt University, Wetenschapspark 7, 3590 Diepenbeek, Belgium; eugene.rameckers@maastrichtuniversity.nl (E.R.); sarah.mingels@kuleuven.be (S.M.); marita.granitzer@uhasselt.be (M.G.); teklugem@yahoo.com (T.G.A.); evi.verbecque@uhasselt.be (E.V.); 3Research School CAPHRI, Department of Rehabilitation Medicine, Maastricht University, 6200 Maastricht, The Netherlands; 4Centre of Expertise in Rehabilitation and Audiology, 6281 Hoensbroek, The Netherlands; 5Musculoskeletal Research Unit, Department of Rehabilitation Sciences, Faculty of Kinesiology and Rehabilitation Sciences, Leuven University, 3000 Leuven, Belgium; 6Department of Special Needs and Inclusive Education, Jimma University, Jimma 378, Ethiopia; 7I-BioStat, Data Science Institute, Hasselt University, 3500 Hasselt, Belgium; liesbeth.bruckers@uhasselt.be

**Keywords:** moderate thinness, PERF-FIT, motor skills, school children, muscular fitness, high-intensity motor learning

## Abstract

Background: Malnutrition has extensive consequences, affecting multiple levels of functioning, including motor skill impairments. However, current interventions have mainly focused on dietary treatment, often neglecting motor impairments and relying solely on clinical and anthropometric indicators to assess treatment response. This study aims to bridge this gap by examining the combined effect of ready-to-use supplementary food (RUSF) and high-intensity motor learning (HiML) on motor skill-related physical fitness in children with moderate thinness (MT). Methods: A cluster randomized controlled trial was conducted among children 5–7 years old with MT in Jimma Town. Three schools were randomized to three intervention arms, including a total of 69 children: RUSF (*n* = 23), RUSF + HiML (*n* = 25), and no intervention (*n* = 21). The HiML training was applied for 12 weeks, and RUSF was distributed daily for 12 weeks. HiML was given daily (1 h/day, 5 days/week). The primary outcome was motor skill-related physical fitness assessed at baseline and endline using the performance and fitness test battery (PERF-FIT). The changes from baseline to endline measurements were calculated as differences, and the mean difference in these changes/differences (DID) was then computed as the outcome measure. AN(C)OVA was used to directly investigate differences between groups. Statistical significance was declared at *p*-value ≤ 0.05. Results: There was a significantly greater and comparable improvement in both the RUSF and RUSF + HiML groups compared to the control group for the ‘stepping’ item (*p* < 0.001), the ‘side jump’ item (*p* < 0.001), the ‘standing long jump’ (*p* < 0.001) and the ‘jumping and hopping’ total (*p* = 0.005). The RUSF + HiML group showed significantly greater improvements in the ‘bounce and catch’ (*p* = 0.001) and ‘throw and catch’ (*p* < 0.001) items compared to the RUSF group, which, in turn, demonstrated greater improvement than the control group in both items (*p* < 0.01). Conclusions: A 12-week combination of RUSF + HiML was proven to be safe in children with MT and caused clear improvements in motor skill-related physical fitness. When the children received RUSF with HiML training, similar gains in stepping, side jump, standing long jump, and jumping and hopping were observed, except for the ball skills where the HiML training group performed better.

## 1. Introduction

Malnutrition, characterized by an imbalance between nutrient intake and requirements, produces energy, protein, or micronutrient deficiencies that hinder children’s growth and development [1]. Malnutrition is a global health issue affecting school children in developing countries [2]. In 2022, 149 million children under five were stunted (low height-for-age), 45 million wasted (low weight-for-height), and 37 million were overweight or obese. Additionally, 190 million children and adolescents aged 5–19 had thinness (low sex-specific BMI-for-age) and 390 million were overweight, with 160 million living with obesity [3]. This has significant implications for their well-being and future prospects of health [2].

Despite the global overnutrition issue, undernutrition in children aged 5–19 remains a pertinent health problem. From 1975 to 2016, global underweight prevalence dropped from 9.2% to 8.4% in girls and 14.8% to 12.4% in boys. However, reductions were limited in low- and middle-income countries (LMICs), with regional rates between 21% and 36% [4]. In Ethiopia, 22% of the school-aged children were still emaciated in 2020 [5].

Good nutrition supports the growth and development of the musculoskeletal and central nervous systems and strengthens the immune system, helping to protect against infections and long-term health issues [6,7]. In contrast, undernutrition exposes children to a higher risk of infections and mortality and increases their likelihood of enduring lasting impacts into adulthood, such as reduced work capacity and greater disease susceptibility [8,9].

Moderate thinness (MT) is a condition where children need nutrient-dense food to support weight gain, height growth, and functional recovery. If there is no intervention, MT can progress to severe thinness (ST), which is life-threatening [10]. Treating MT requires high-energy intake and essential nutrients to address deficiencies and promote normal growth [11,12], for which supplementation is a frequently applied and successful approach. Ready-to-use supplementary food (RUSF) is a lipid-nutrient spread designed to provide concentrated energy and micronutrients for children with MT. RUSF is energy-dense, resistant to bacterial contamination, and does not require end-user preparation, making it effective in treating MT [13]. While lipid-based RUSF halts the progression of MT to ST [14] and improves anthropometric measures [15], body composition, and isometric muscle strength [16], its effects on motor skill-related physical fitness in MT children have not yet been established.

Due to its far-reaching impact on overall functioning, malnutrition affects not only children’s nutritional status, body composition [17], and muscle strength but also their motor skill-related physical fitness [18,19]. This motor skill-related physical fitness refers to the neuromuscular components of fitness that enable a child to successfully perform a motor skill, game, or activity [20]. It includes agility, balance, coordination, power, reaction time, and speed, all of which are essential for daily physical needs and peer participation [21]. Even though motor skill problems in undernourished children are rarely reported, the available research indicates contradictory results. Some studies report clear motor difficulties [22,23,24,25,26], whereas others do not. Such contradictions seem to depend on the sample under investigation. Assessing and training motor skills in early childhood is crucial, as these skills directly affect daily activities, school sports, and leisure while also fostering an active lifestyle into adulthood. The early years, from birth to age 8, are particularly vital for a child’s physical, cognitive, and socioemotional development [26]. This developmental period is especially important between the ages of 5 and 7 when children show a marked improvement in mastering various motor skills [27]. However, during this crucial period, children are also vulnerable to disease and malnutrition. Rapid growth [28] at this stage increases the need for age-appropriate physical fitness and motor skills, which are essential for engaging in physical activities, exploration, and developing social skills [29]. These developmental needs can be hindered if children experience wasting [30].

According to World Health Organization (WHO) guidelines, children and adolescents aged 5 to 17 benefit from engaging in moderate-to-vigorous physical activity (MVPA) for an average of 60 min per day [31]. These guidelines can be easily achieved when children engage in games, sports, housework, recreation, physical education, or planned exercises as part of family and school activities [32]. Nevertheless, many youths fail to meet this recommendation [33,34], underscoring the need to enhance fitness during childhood and adolescence to address physical inactivity and reduce disparities in activity levels as they grow older [35,36]. Additionally, muscle weakness—a common consequence of undernourishment in children with moderate thinness (MT)—can result in functional limitations, injuries, and poor health outcomes [37]. Targeted interventions are essential for improving motor skill-related physical fitness and altering physical activity patterns [38,39]. Research shows that regular participation in youth resistance training can significantly enhance fundamental movement skills such as jumping, throwing, and running [40]. There is a positive association between muscular fitness and physical activity, particularly vigorous physical activity, in children and adolescents [41]. However, nearly 90% of children with low motor competence fail to meet the public health recommendations [42].

The pediatric activity pyramid (PAP) emphasizes the importance of strength and skill activities in promoting physical activity for children and adolescents [43]. Physical fitness includes attributes such as cardiorespiratory endurance, muscular strength, flexibility, and body composition [44]. Motor skills involve coordinated movement patterns for everyday tasks like running, jumping, throwing, catching, and balancing [45]. Both strength and motor skill activities are crucial for overall physical fitness and well-being and tend to be implicated in children with MT.

While stand-alone free play is insufficient, active play addresses the entire PAP. Active play engages all muscle groups, promotes physical activity, fun, and enhances gross motor skills [46]. Schools could serve as a perfect place to promote and organize active play; however, many teachers lack the skills to develop specific programs targeting the entire PAP. Further, there is a lack of structured movement programs to stimulate active play, especially in disadvantaged communities such as those in Ethiopia. Therapeutically (physical/occupational), stimulating this MVPA daily through active play, may be considered a form of high-intensity motor learning training (HiML). It is highly intensive because it is performed daily for a longer period (minimum volume: 30–40 h of training) with the focus of a high time on task during training (training/rest ratio minimally 70–30%). The term ‘motor learning’ refers to the training of motor skills [47]. Such training modalities (e.g., HABIT-ILE) are already highly effective in improving motor skill competence in children aged five and above with developmental impairments such as cerebral palsy [48,49] and are currently recommended in therapeutic settings. Notwithstanding, the effect of these approaches on MT children is unknown.

No previous studies have assessed the effect of RUSF or its combination with HiML on motor skill competence in MT children (ages 5–7). According to Gallahue’s hourglass model of motor development theory, children learn motor skills sequentially. The optimal age for fundamental movement skill (FMS) development is 5–7 years [50,51]. Children aged five to six years old undergo a crucial developmental stage, which is also the optimal age for motor skill development [52]. It is not only age that affects motor skill competence. A systematic review and meta-analysis revealed gender differences in FMS levels in children aged 3–6 years. Boys outperformed girls in object control skills and locomotor skills, and this difference increased with age [53]. Given the combined effects of HTML and RUSF on children with moderate thinness and the lack of publications addressing how moderate thinness impacts gross motor development, muscular fitness, and motor skills—both globally and specifically in Ethiopia—it is essential to conduct a more in-depth analysis of this issue. This study hypothesized that MT children 5–7 years of age who received dietary supplementation (RUSF) alone or combined with functional high-intensity motor training have better motor skills and, muscular fitness compared to those who receive neither. Therefore, this study primarily investigated the effect of RUSF with(out) HiML compared to no intervention on motor skill-related physical fitness in 5–7-year-old MT children. Additionally, it was examined whether the effect of the intervention differed according to age and sex.

## 2. Methods

### 2.1. Study Area and Period

The study was conducted in kindergartens and primary schools located in the Jimma Town of the Oromia Region in Southwestern Ethiopia, which is situated 357 km from Addis Ababa, the capital city. For the study, three schools were selected: Mendera, Jiren, and Dilfere. Data were collected at baseline and end line, with an intervention in between. Baseline data collection occurred from 7 June to 30 June 2023; the intervention ran from 5 July to 5 October 2023; and endline data were collected from 10 October to 30 October 2023.

### 2.2. Study Design

A cluster-randomized controlled trial with parallel groups in a three-arm study design was conducted. The three arms included: MT children receiving RUSF (Arm 1), MT children receiving RUSF + HiML training (Arm 2), and a control group, where the MT children received no intervention (Arm 3). Three schools were selected from all the schools in the study area. Each school was randomly assigned to one of three study groups: two intervention groups and one control group. Within each selected school, eligible children were randomly chosen until the target number of participants (20–25 children per school) was reached. The trial followed the CONSORT recommendations for cluster-randomized trials [54]. Participation of the children required informed consent from the children’s mothers or caregivers. The random assignment was carried out by randomly drawing three clusters from a bag. The drawn clusters were then alternately assigned to the two interventions or control groups (Figure 1). This random allocation was managed by a blinded team member. The trial was registered (PACTR202305718679999) at the Pan African Clinical Trials Registry (PACTR). Ethical approval was obtained (JUIH/IRB/324/23).

### 2.3. Participants

#### Selection Criteria

Children were considered for inclusion if they presented with MT ((BMI-for-age ≥−3 to <−2), and were aged 5 to 7 years. Children were excluded if they were on an outpatient therapeutic program (OTP), had overt physical disabilities, visual/auditory problems, tuberculosis (TB), HIV/AIDS, allergies to supplement ingredients, skin infections, plans to leave the study area within the next 6 months, serious illnesses, bilateral pitting edema, walking impairment due to orthopedic reasons, illness or medical complications that prevent the safe consumption of supplementary food, or were participating in other interventions.

### 2.4. Sample Size Determination and Sampling Technique

In the absence of prior studies to guide the sample size calculation, we assumed a medium effect size of 0.5, a design effect of 1.5 to account for clustering at the school level, a 10% loss to follow-up, and a significance level of 5%. Using G*Power 3.1.9.4, the required sample size for an ANOVA with three arms and 80% power was estimated at 42 participants. After adjusting for the design effect and expected loss to follow-up, the final sample size was calculated to be 69.

The schools were selected using a simple random sampling technique. The respective total sample was allocated to each school using a proportion of the number of children in the respective age for each school. Within the schools, children were randomly selected to be screened for thinness (based on BMI-for-age) from the eligible participants using their school registration book as a sampling frame. Then, the screened children were stratified by MT and WN and selected by a simple random sampling technique using their registration number as a sampling frame.

### 2.5. Interventions

The children in the RUSF group and the RUSF + HiML group received one sachet per day every day of the week (7 sachets per child per week) at school, for 12 weeks. The RUSF was provided in a quantity sufficient to meet their nutrient requirements (500 kcal/day). Each energy bar contained 12.5 g proteins, 31 g fat, and 42.8 g carbohydrates. The children in the RUSF group received the supplement directly in the morning. The children in the RUSF + HiML group received the supplement as breakfast after 60 min of HiML training.

The children in the RUSF + HiML group received active play for 60 min per day, five days a week for 12 weeks (Appendix A). The intensity of this training was within the duration of the intervention, and the total training duration/volume of hours on the task activity was 60 h. The HiML training was given by three sports science professionals. Monitoring was implemented to ensure that the supplementary foods were consumed only by the enrolled school children. The children in the control group did not receive any dietary or HiML training intervention. Since the trial was run during the summer holidays, the children in the control group stayed at home.

### 2.6. Data Collection

After enrollment, the selection criteria were checked by measuring the child’s weight and height. Trained data collectors, with assistance from health extension workers, conducted the initial screening through observation based on exclusion criteria. This was followed by a further objective screening that involved measuring BMI-for-age and checking for bilateral pitting edema. The screening process continued per school until the target of 69 participants was reached (20–25 children per school). Children who met the BMI-for-age criteria then participated in the baseline data collection at the school. Through face-to-face interviews using a structured questionnaire, parents or caregivers provided information about the child’s health, feeding practices, immunization history, and family sociodemographic characteristics. These interviews, along with the child’s measurements, were conducted at the school.

Since the effects of the household wealth index and household food security were considered in the analysis, indices were established for both factors.

The household wealth index was determined using principal component analysis (PCA) based on 27 items related to household assets. Several assumptions were checked during the PCA process, including the ratio of cases to variables (18:1), correlation matrix, anti-image values, Kaiser–Meyer–Olkin (KMO) measure, and Bartlett test of sphericity. After ensuring that all assumptions were met, the resulting scores were ranked and classified as poor, medium, and rich [55].

Household food insecurity was determined using the Household Food Insecurity Access Scale (HFIAS), developed by the Food and Nutrition Technical Assistance (FANTA) project and validated in Ethiopia [56,57]. The scale assesses the food security status based on nine questions about the occurrence and frequency of food insecurity. The data were analyzed using specific classification criteria to categorize households as food-secure and food-insecure.

### 2.7. Data Quality Control

Data were collected by one physiotherapist, three nurses, and two nutritionists, who also supervised the process. All data collectors received four days of basic skill training on isometric muscle strength measurements and the PERF-FIT test battery administration. Daily quality checks were conducted to ensure data accuracy.

#### 2.7.1. Anthropometric Measurements

The same person conducted all anthropometric measurements to ensure consistency and avoid interobserver variations.

Height (cm) was measured using a portable stadiometer, the Seca 213, manufactured by Seca in Hamburg, Germany. Children were instructed to stand barefoot against a wall, with their hair pulled back and eyes looking straight ahead (Frankfurt plane) to ensure their line of sight was perpendicular to the stadiometer. They were told to keep their knees straight, while their heels, calves, buttocks, and shoulder blades touched the stadiometer. The height measurement was recorded to the nearest 0.1 cm [58].

Body weight (kg) was measured using a Seca digital weighing scale (Model 770), manufactured by Seca in Hamburg, Germany. Before each measurement, the scale was calibrated to ensure a zero reading, and it was validated daily using a 1 kg standard weight. Children were weighed barefoot and with light clothing. The measurement was recorded to the nearest 0.1 kg [59].

#### 2.7.2. Performance and Fitness Test Battery

The performance and fitness (PERF-FIT) test battery is an assessment tool designed to measure motor skill-related physical fitness in children aged 5 to 12 years residing in low-income settings [60,61]. The PERF-FIT has been proven to be reliable, valid, and applicable in diverse resource-limited environments [61,62,63]. It assesses fundamental motor skills and musculoskeletal fitness. The PER-FIT consists of two subscales: the motor skill performance subscale, which consists of five skill item series (SIS), and the agility and power subscale, which includes five items.

The motor skill performance subscale contains a series of five tasks with increasing difficulty, known as SIS: (a) bounce and catch (count, #), (b) throw and catch (count, #), (c) static balance (s), (d) dynamic balance (count, #), (e) jumping and hopping (count, #). Children start at the easiest level of the SIS. If children complete the first trial without making any mistakes, no second trial is given, and they move on to the next level of the task. A discontinuation rule is in place. If a child is unable to score more than half of the maximum points in either of the two trials for a SIS item, that series is discontinued, and the next level of that SIS is not administered.

The agility and power subscale consists of 5 items designed to assess aspects of anaerobic capacity, including explosive power, running, and agility. These items are the standing long jump, overhand throw heavy bag, running, stepping, and jumping. All power and agility tasks comprise two independent test trials, regardless of the score obtained. Children were given at least 15 s of rest between the two test attempts (Appendix A).

### 2.8. Statistical Analysis

The data were doubly entered into Epi Data 3.1 and exported to Statistical Package for Social Sciences (SPSS) version 29.0 for cleaning and analyses. The collected data were checked, cleaned, and coded to avoid inconsistencies and incompleteness before analysis. Incomplete and inconsistent data were excluded from the analysis. To explore the distribution of sociodemographic variables between the study arms, a Chi-Pearson square was calculated. Descriptive statistics are presented as mean ± SD. Analysis of variance (ANOVA) was performed to determine the difference in differences (DID) between the intervention arms. Statistically non-significant factors and/or interaction effects were removed from the model to increase its power. Assumptions for applying ANOVA were checked. First, the residuals’ normality was checked with skewness and kurtosis z-values (reference: between −1.96 and +1.96), the Shapiro–Wilk test (*p*-value > 0.05), and visual inspection of the histogram, box plots, and the normal Q-Q plots. Levene’s test was used to assess homoscedasticity. In case of significant differences between the groups, an analysis of covariance (ANCOVA) was run where the baseline results of the outcome were added as a covariate [23,64] and reported if significant. The final model was used to explore pairwise comparisons between the groups, using LSD. In cases where the assumptions for ANOVA were not met, a similar non-parametric generalized linear model (generalized estimating equations model) was conducted using Wald chi-square statistics. Statistical significance was set at alpha less than 0.05.

## 3. Results

### 3.1. Participants

Table 1 provides an overview of the sociodemographic characteristics of the parents/caregivers, families, and children. Children in the RUSF + HiML group are more likely to live in larger families, with 48% coming from families of five or more members, compared to approximately 9% in both the control and RUSF groups. Vitamin A supplementation is less common in the RUSF + HiML group, where only 24% of the children receive supplements, compared to 29% and 22% in the control and RUSF groups, respectively. Complementary feeding is introduced later in the control group, with 62% receiving it after six months, compared to 26% and 28% in the RUSF and RUSF + HiML groups. No other significant differences in sociodemographic characteristics were observed between the groups.

### 3.2. Intervention Effects on Skill-Related Physical Fitness

At baseline, there were no differences between the groups in their performances on each of the PERF-FIT items (Appendix A).

The study compared the differences between the baseline and endline values. Significant differences were found in the stepping total, side jump total, long jump total, overhand throw total, bounce and catch total, and throw and catch total. However, the running total did not show significant differences between both arms. The results indicate a statistically significant inside jump total with both arms: RUSF + HiML had a mean of 4.56 ± 3.44, RUSF 2.66 ± 2.31, and the no intervention arm 0.71 ± 2.00 (*p* < 0.001). The long jump total was 19.61 ± 23.13 (*p* < 0.001), and the overhand throw total was 19.30 ± 25.29 (*p* = 0.05). Additionally, there was a significant difference in bounce and catch total (9.55 ± 12.66, *p* < 0.001) and throw and catch total (7.03 ± 13.47, *p* < 0.001) between the intervention (RUSF and RUSF + HiML groups) and no intervention groups (Appendix A).

The DID for each PERF-FIT variable is presented in Table 2 and the items showing between group differences visualized in Figure 2 and Figure 3. Significant intervention effects were found for three out of five agility and power subscale items: stepping (F_2,46_ = 27.579, *p* < 0.001, η^2^ = 0.545), side jump (F_2,63_ = 13.995, *p* < 0.001, η^2^ = 0.308), and standing long jump (F_2,65_ = 14.072, *p* < 0.001, η^2^ = 0.305) (Figure 2). There was a significant association with the intervention effects and the baseline performance for the side jump and the standing long jump (Table 2). For each variable, the post hoc comparison revealed that both the RUSF and RUSF + HiML groups presented with significantly greater improvement (larger DID) after the intervention compared to the control group. No differences were found between both intervention groups for these variables, except for the side jump where the RUSF + HiML group improved more than the RUSF group (Table 2).

Significant intervention effects were found for three out of five motor skill performance subscale SIS: bouncing (F_2,63_ = 59.243, *p* < 0.001, η^2^ = 0.653), throwing (F_2,63_ = 40.216, *p* < 0.001, η^2^ = 0.561), and jumping and hopping (W (2) 44.110, *p* = 0.005) (Figure 3). There was a significant association with the intervention effects and the baseline performance for all these variables (Table 2). Specifically, for jumping and hopping, a significant interaction effect was found with sex (W (2) = 11.438, *p* = 0.010). The largest performance improvements were seen for bouncing and throwing in the RUSF + HiML group. These children outperformed both those in the RUSF and control groups (*p* ≤ 0.001, Table 2). The RUSF group also performed significantly better after the intervention compared to the control group (Table 2). The DIDs for each significantly changed SIS are illustrated in Figure 3. The DID for jumping and hopping SIS was only significant for the RUSF and RUSF + HiML groups compared to the control group; no differences were found between the two intervention groups for this variable post hoc.

## 4. Discussion

This study aimed to investigate the effects of ready-to-use supplementary food (RUSF) with or without high-intensity motor learning (HiML) compared to no intervention on motor skill-related physical fitness in children with moderate thinness (MT), as well as the potential roles of sex and age in their recovery. The study results revealed that after 12 weeks of intervention, children with MT who received RUSF with(out) HiML significantly improved in various physical activities such as stepping, side jumps, standing long jumps, bouncing and catching, throwing and catching, and jumping and hopping compared to those who received no intervention. For most activities, both RUSF groups improved similarly. However, the side jump and the ball skills (bounce and catch, throw and catch) improved significantly more in the RUSF + HiML group compared to the children who received only RUSF or no intervention. No significant improvements were observed in running, static balance, and dynamic balance performances between the groups. Interestingly, the RUSF group without HiML presented with the largest improvements for the standing long jump and with similar improvements to the RUSF + HiML group on the jumping and hopping SIS. Although age and sex play a role in the acquisition of motor skill-related physical fitness, these variables did not seem to influence the children’s recovery, as evidenced by the lack of interaction effects except for the jumping and hopping SIS.

### 4.1. Impact of RUSF

Whether or not the children additionally received HiML, their muscular fitness and jumping and hopping performances increased significantly compared to when receiving no intervention. Agility, coordination, and explosive lower limb power are essential for the items that were responsive to the supplementation. Particularly, the power component requires adequate muscle function, which is known to be impeded by undernourishment [16,65]. We already established that under RUSF, muscle strength of the lower legs significantly increased [16], which may explain the progress in these specific activities. Our study is the first to explore the impact of supplementation on motor skill-related fitness, which is beyond the nutrition-related outcomes commonly reported in the literature [11,13,15,66,67]. Although the children with MT represent a distinct group of undernourished children, treatment guidelines are lacking. Our results indicate that providing them with supplementation improves not only their nutritional status and isometric muscle strength [16], but also their lower limb muscular fitness, which is key to overall functioning. Nevertheless, some items seem to be more responsive to actual skill training.

### 4.2. Impact of High-Intensity Motor Learning Training

A recent systematic review suggests that motor skill training interventions significantly improve motor skill development in typically developing children [68] and other pediatric populations, such as children with CP, developmental coordination disorder (DCD), and even children with overweight or obesity [48,49,50,51,69,70,71,72,73]. Despite their clear potential in a rehabilitation context, the effect of motor skill interventions on children suffering from MT has not been tested so far. This can probably be explained by the fear of further breaking down the energy level and wasting muscle mass through training in this specific group of children. However, it seems that active play, a fun way of training motor skills, combined with supplementation (RUSF) is not only safe but even stimulates the progression of motor skill-related physical fitness in these children. Interestingly, similar improvements in muscular fitness were discovered, showing that the higher calorie demands during skill training did not result in reduced gains. This type of play entails repeating activities that focus on a specific skill set over a series of training sessions. The activities are designed to be enjoyable and engaging for children, encouraging them to move in various ways. When children find the activities fun, they are more likely to be motivated to perform repetitive motions or activities consistently over an extended period. Children may not recognize that engaging in these activities is a form of muscle training to improve their motor skills. These active play interventions involving cooperative games can therefore be used to improve FMS and promote health and fitness benefits, ultimately increasing the physical activity levels of school-aged children (5–7 years old) [74].

Furthermore, active play involves repetitive muscle movements to build strength, speed, and agility, leading to the development of gross and fine motor skills [69]. The available literature on active play or goal-oriented play among preschoolers and/or early school-aged children shows improvements in ball skills, balance, walking, running, and jumping/hopping [51,69,71,74]. This is not entirely in line with the findings of the current study. Although the intervention duration was similar (12 weeks), the intensity was one hour per day and we focused on locomotor skills/gross motor, playground/ball skills, and cultural play activities. We did find clear advantages of HiML training for the ball skills (trained), but no larger changes were seen in running in an agility ladder (untrained), static balance (untrained), dynamic balance (untrained), and jumping/hopping (untrained) compared to the other intervention groups. Despite the slightly different effects on the types of motor skills, the available studies have shown that high-intensity training modalities such as Hand–Arm Bimanual Intensive Training Including Lower Extremity (HABIT-ILE), and Functional Intensive Treatment (FITCARE) are highly interconnected, motor-based, and proved to be effective in improving developmental domains such as motor skills, cognitive abilities, and language development in children aged five and older with various types of developmental impairments [48,49]. Regular muscle activity improves blood circulation, which helps deliver the necessary substances to the nervous system and muscles. The impact of fulfilled nutrients on the nervous system and muscles enhances fitness by improving speed, endurance, balance, and coordination, all of which are vital for motor ability. This could explain why children who received RUSF + HiML improved significantly more on their side jump performance, an item that requires agility, coordination and speed. From that point of view, our results add to the body of knowledge regarding the feasibility and effectiveness of this type of training in children with MT.

### 4.3. Impact of Age and Sex

The present study revealed that male participants in the combined RUSF + HiML group gave a better jumping and hopping performance compared to both the RUSF and no-intervention groups, particularly in females. No other such differences were identified in our sample. Furthermore, no interaction effects with age were seen either across all PERF-FIT outcomes. Hence, despite the well-established impact of age and the potential effects of sex on motor skill-related performances in preschool children [27,75,76,77,78], both sexes regardless of their age group responded similarly to the training. The overall lack of impact of sex and age on the treatment effect is consistent with a study conducted in Tehran, Iran, that found that a 12-week motor activity program had a significant impact on the dynamic balance, motor speed, and strength in both boys and girls aged 5 to 6 [3]. However, the activities did not significantly affect the coordination of the girls [79]. In the southwestern United States, following an 8-week, need-supportive, FMS-based afterschool program in public elementary schools, both boys and girls aged 5 to 8 years old in the intervention group showed greater improvements in FMS competence compared to those in the control group [80]. Thus, regardless of their potential baseline differences, they responded similarly to the training program and supplementation. The study conducted in Ethiopia demonstrates that play-based psychomotor and psychosocial stimulation positively impacts the motor development (both fine and gross) of children with severe acute malnutrition (SAM) aged 6 months to 6 years. This research indicates that play-based stimulation is beneficial for treating SAM children under six in low-income settings. The findings revealed a significant improvement in gross motor functions during the hospital stay, while fine motor functions showed improvement after discharge during home follow-up. Both younger and older children reaped similar benefits from the intervention [81]. Additionally, a study conducted in South Africa found that moderate levels of stunting and wasting negatively affected school performance and motor functioning in school beginners [23]. A study conducted in South Africa indicated that moderate to vigorous physical activity levels were positively associated with health-related physical fitness (HRPF), motor-related physical fitness (MRPF), and certain motor skills in children aged 5 to 8 years [82]. The recent research on the same age group indicated that the combination of RUSF and HiML interventions improved the body composition, height, weight, and muscle strength of the moderately thin children studied [16].

### 4.4. Strengths and Limitations of the Study

Although multiple studies investigated the effect of motor skill training among healthy children and children with severe motor disorders, our study is the first to apply this training approach to children who suffer from MT. A cluster randomization was implemented to prevent information contamination. It is important to note that the children’s age, ranging from 5 to 7 years, could be considered as a limitation because it may not show much variation and therefore restrict the generalizability of the results to younger and older children. The study’s primary limitation is that the children’s dietary intake during the intervention was not monitored. This study has some limitations that must be acknowledged. One potential issue is social desirability bias, which was minimized by informing respondents that their answers were for comparison purposes only and would not affect service use or their privacy. Additionally, it is important to note that the children’s dietary intake during the intervention was not monitored. Furthermore, due to the fact that we intended to avoid contamination between participants, we aimed to conduct a cluster RCT. However, since there is only one school per intervention arm, the current results should be interpreted with caution. The design as it is, despite its random selection of children within a cluster, may have caused selection bias. The sample size does not provide adequate statistical power for this specific design, thus increasing the risk of type I errors, and may have also caused residual and unmeasured confounding. Larger studies with multiple clusters per intervention arm are needed to verify the results reported in this exploratory pilot cluster RCT.

### 4.5. Recommendations for Future Research and Clinical Practice

The findings suggest that kindergartens, schools, and communities should prioritize scientifically rigorous physical education programs, such as this HiML training, which is a form of active play consisting of goal-oriented play activities. The HiML approach can help preschool children with MT develop their muscular fitness and motor skills which are essential for timely motor development. The current study therefore provides policymakers and guideline developers with a rationale for implementing motor skill interventions into the school curriculum. Physical education classes offering a variety of skills daily can play an essential role in supporting and promoting the healthy development of preschool children.

In the current study, the children received supplements to ensure their calorie uptake. However, other ways of securing healthy foods should be explored to empower families and their children in the future. For instance, in Ethiopia, dietary habits may be monotonous because of a lack of knowledge and not because of food insecurity per se. In such cases, where there is sufficient access to food, the potential role of nutritional education should be explored in the future. In insecure food areas, on the other hand, solutions providing the children with sufficient energy need further attention as a lack not only hurts their nutritional status but also induces far-reaching consequences that transcend their physical capacities. This would help us gain more insight into why motor competence is low; the impact of inequalities on motor competence; and whether there are specific subgroups that need interventions to improve the motor competence of children with MT.

## 5. Conclusions

The study demonstrated that a 12-week combination of RUSF + HiML is safe in children (5 to 7 years) with MT and caused clear improvements in motor skill-related physical fitness. When the children received RUSF with HiML training, similar gains in stepping, standing long jump, and jumping and hopping were identified, except for the side jump and ball skills where the HiML training group performed better.

This study provided evidence that HiML training with a combination of RUSF can be used to boost motor development in this age group, resulting in significant progress in children with MT. Based on these findings, we suggest expanding the HiML training program to kindergartens, schools, and communities to promote healthy development across childhood.

The findings of this study will assist policymakers, both governmental and non-governmental, in Ethiopia. This will enable them to implement appropriate policies to combat malnutrition effectively. Investing in and testing alternative interventions and conducting further research in schools in rural and socioeconomically disadvantaged areas is worthwhile, as these may provide solutions to the questions raised by the current study.

## Figures and Tables

**Figure 1 nutrients-17-00030-f001:**
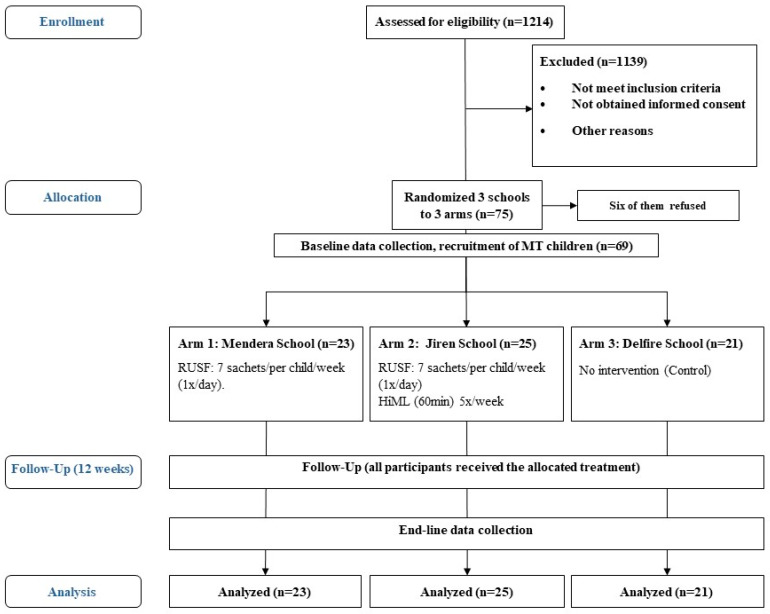
Study flow chart, 2023 (*n* = number of children, MT = moderate thinness, HiML = high-intensity motor learning, RUSF = ready-to-use supplementary food).

**Figure 2 nutrients-17-00030-f002:**
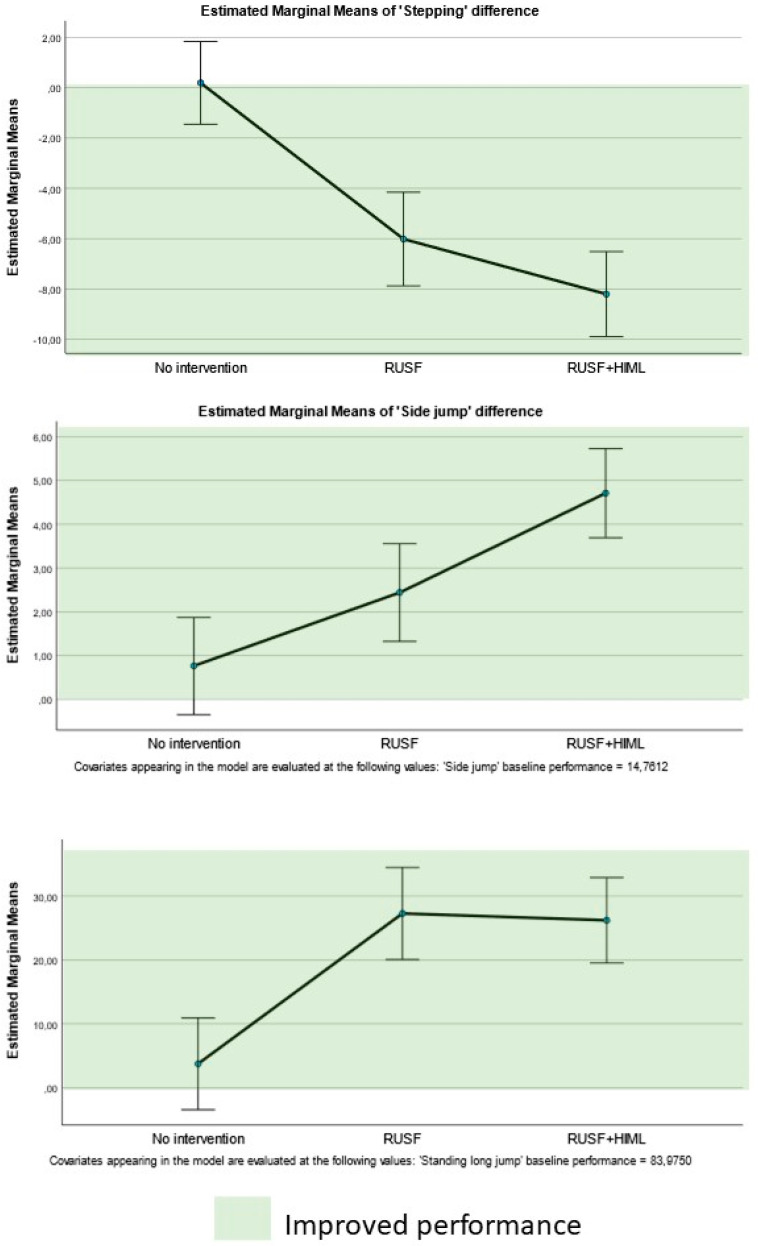
Comparisons of the differences pre–post per type of intervention for the muscular fitness items. Legend: the green zones in each graph indicate the range of improvement (i.e., better performance).

**Figure 3 nutrients-17-00030-f003:**
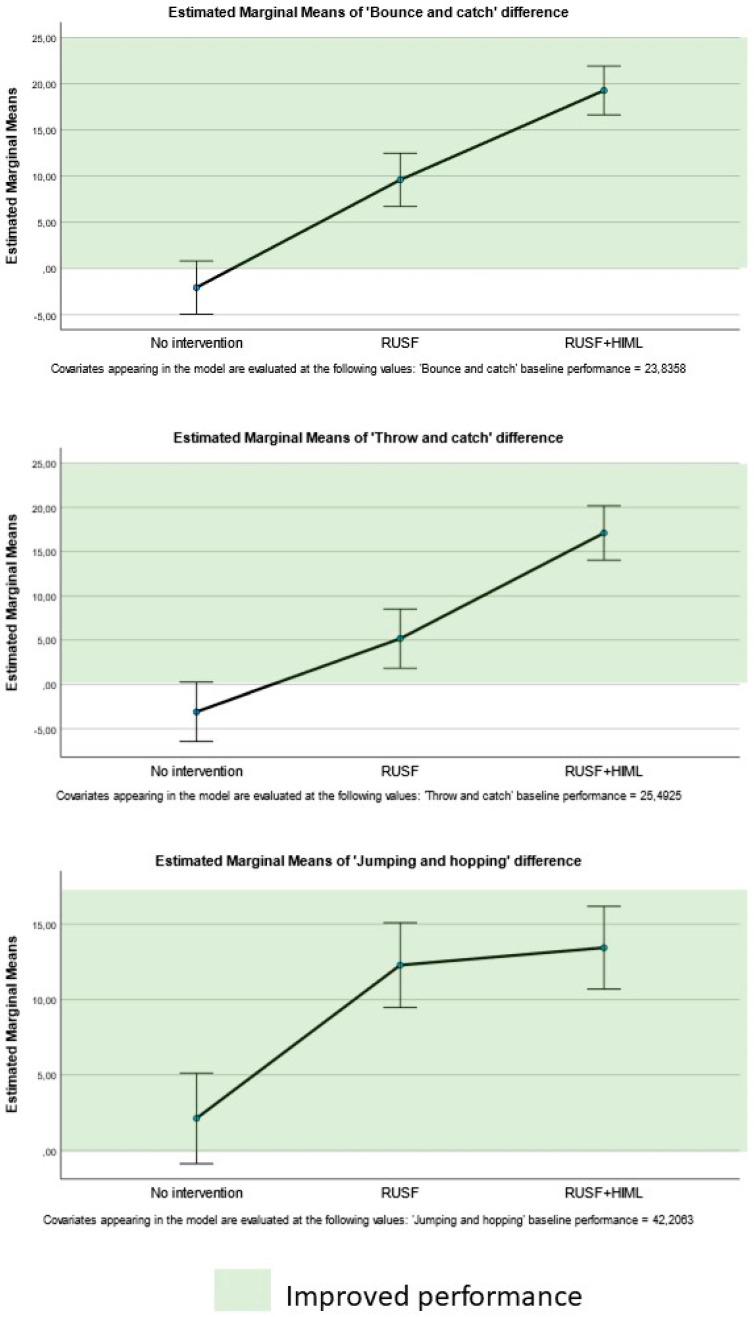
Comparisons of the differences pre–post per type of intervention for the skill item series.

**Table 1 nutrients-17-00030-t001:** Sociodemographic characteristics of the parents/caregivers and children undergoing intervention in Jimma Town, Southwest Ethiopia, 2023 (*n* = 69).

Sociodemo-Graphic Variables	Category	Intervention Groups								
		Control (*n* = 21)		RUSF (*n* = 23)		RUSF + HIML (*n* = 25)		Total		*p*
		N	%	N	%	N	%	N	%	
Parents/caregivers and family										
Marital status of the caregiver	Married	18	85.7	18	78.3	20	80.0	56	81.2	0.270
	Divorced	3	14.3	1	4.3	3	12.0	7	10.1	
	Widowed	0	0.0	1	4.3	1	4.0	2	2.9	
	Separated	0	0.0	0	0.0	1	4.0	1	1.4	
	Single	0	0.0	3	13.0	0	0.0	3	4.3	
Age (years) of mother/caregiver	<30	13	61.9	11	47.8	16	64.0	40	58.0	0.478
	≥30	8	38.1	12	52.2	9	36.0	29	42.0	
Family size	<5	19	90.5	21	91.3	13	52.0	53	76.8	0.001
	≥5	2	9.5	2	8.7	12	48.0	16	23.2	
Educational status of the mother	Cannot read and write	2	9.5	5	21.7	6	24.0	13	18.8	0.956
	Can read and write	2	9.5	3	13.0	3	12.0	8	11.6	
	Primary (0–8)	10	47.6	8	34.8	9	36.0	27	39.1	
	Secondary (9–12)	5	23.8	5	21.7	4	16.0	14	20.3	
	Above secondary (>12)	2	9.5	2	8.7	3	12.0	7	10.1	
Occupation of mother	Housewife	12	57.1	7	30.4	9	36.0	28	40.6	0.658
	Merchant	0	0.0	1	4.3	2	8.0	3	4.3	
	Gov’t employee	5	23.8	6	26.1	6	24.0	17	24.6	
	Self-employed	3	14.3	7	30.4	5	20.0	15	21.7	
	Other (daily laborer, wood seller)	1	4.8	2	8.7	3	12.0	6	8.7	
Wealth index	Poor	3	14.3	10	43.5	8	32.0	21	30.4	0.237
	Medium	16	76.2	11	47.8	1	52.0	40	58.0	
	Rich	2	9.5	2	8.7	4	16.0	8	11.6	
HFIA category	Severe food insecurity	21	100.0	23	100.0	25	100.0	69	100.0	/
Child										
Sex at birth	Male	10	47.6	11	47.8	10	40.0	31	44.9	0.825
	Female	11	52.4	12	52.2	15	60.0	38	55.1	
Age group	5	6	28.6	7	30.4	4	16.0	17	24.6	0.180
	6	2	9.5	2	8.7	8	32.0	12	17.4	
	7	13	61.9	14	60.9	13	52.0	50	58.0	
Grade level	Zero grade	8	38.1	8	34.8	12	48.0	28	40.6	0.623
	Grade one	13	61.9	15	65.2	13	52.0	41	59.4	
Child fully immunized	Yes	19	90.5	22	95.7	24	96.0	65	94.2	0.680
	No	2	9.5	1	4.3	1	4.0	4	5.8	
Deworming tablet in the last 6 months	Yes	15	71.4	18	78.3	13	52.0	46	66.7	0.134
	No	6	28.6	5	21.7	12	48.0	23	33.3	
A child gets vitamin A supplementation	Yes	15	71.4	18	78.3	6	24.0	39	56.5	<0.001
	No	6	28.6	5	21.7	19	76.0	30	43.5	
EBF for the first 6 months	Yes	14	66.7	10	43.5	15	60.0	39	56.5	0.273
	No	7	33.3	13	56.5	10	40.0	30	43.5	
Complementary feeding	Before 6 months	7	33.3	13	56.5	10	40.0	30	43.5	0.030
	At 6 months	1	4.8	4	17.4	8	32.0	13	18.8	
	After 6 months	13	61.9	6	26.1	7	28.0	26	37.7	

Abbreviations: HFIA: Household Food Insecurity Access Scale, EBF: exclusive breastfeeding, significant at *p* < 0.05.

**Table 2 nutrients-17-00030-t002:** The differences in difference of the effect of the intervention on the dependent variables for their arm of the study (post hoc tests) among 5–7 years of age MT children in Jimma Town, Southwest Ethiopia, 2023.

Variables	Intervention Type	N	Difference in Baseline–Endline Difference (DID)	Main Effect	Post Hoc Comparison	
			Mean ± SD/SE ^$^ Median ± IQR			
Running total (seconds)	No intervention	21	−1.05 ± 1.60	F_2,35_ = 1.511, *p* = 0.235, η^2^ = 0.079	/	
	RUSF	16	−0.42 ± 4.25			
	RUSF + HiML	13	−2.59 ± 2.36			
	Total	50	−1.20 ± 2.96			
Stepping total (seconds)	No intervention	21	0.19 ± 3.28	F_2,46_ = 27.579, *p* < 0.001, η^2^ = 0.545	RUSF + HiML = RUSF RUSF + HiML > control RUSF > control	*p* = 0.226 *p* < 0.001 *p* < 0.001
	RUSF	18	−6.01 ± 3.67			
	RUSF + HiML	19	−8.20 ± 3.47			
Side jump total (# jumps) ^$^	Baseline performance		Mean: 14.76	F_1,63_ = 10.162, *p* = 0.002, η^2^ = 0.139		
	No intervention	21	0.76 ± 0.55	F_2,63_ = 13.995, *p* < 0.001 η^2^ = 0.308	RUSF + HiML > RUSF RUSF + HiML > control RUSF > control	*p* = 0.004 *p* < 0.001 *p* = 0.037
	RUSF	21	2.44 ± 0.56			
	RUSF + HiML	25	4.71 ± 0.51			
Long jump total (cm) ^$^	Baseline performance		Mean: 83.98	F_1,64_ = 32.823, *p* < 0.001, η^2^ = 0.339		
	No intervention	21	3.74 ± 3.60	F_2,65_ = 14.072, *p* < 0.001, η^2^ = 0.305	RUSF + HiML = RUSF RUSF + HiML > control RUSF> control	*p* = 0.832 *p* < 0.001 *p* < 0.001
	RUSF	22	27.28 ± 3.61			
	RUSF + HiML	25	26.21 ± 3.35			
Overhand throw total (cm)	No intervention	21	8.47 ± 11.94	F_2,62_ = 2.245, *p* = 0.115, η^2^ = 0.089	/	
	RUSF	22	22.12 ± 30.14			
	RUSF + HiML	25	26.04 ± 26.95			
Bounce and catch total (points/50) ^$^	Baseline performance		Mean: 23.84	F_1,63_ = 47.42, *p* < 0.001, η^2^ = 0.429		
	No intervention	21	−2.06 ± 1.44	F_2,63_ = 59.243, *p* < 0.001, η^2^ = 0.653	RUSF + HiML > RUSF RUSF + HiML> control RUSF > control	*p* < 0.001 *p* < 0.001 *p* < 0.001
	RUSF	22	9.60 ± 1.44			
	RUSF + HiML	25	19.27 ± 1.32			
Throw and catch total (points/50) ^$^	Baseline performance		Mean: 25.49	F_1,63_ = 48.913, *p* < 0.001, η^2^ = 0.437		
	No intervention	21	−3.09 ± 1.67	F_2,63_ = 40.216, *p* < 0.001, η^2^ = 0.561	RUSF + HiML > RUSF RUSF + HiML> control RUSF > control	*p* < 0.001 *p* < 0.001 *p* < 0.001
	RUSF	22	5.16 ± 1.67			
	RUSF + HiML	25	17.10 ± 1.54			
Static balance (seconds) *	No intervention	21	3.00 ± 5.63	W(2) = 1.475, *p* = 0.478	/	
	RUSF	22	0.00 ± 10.00			
	RUSF + HiML	25	0.00 ± 6.00			
Dynamic balance (points/32) *	No intervention	20	4.50 ± 4.00	W(2) = 0.186, *p* = 0.911	/	
	RUSF	21	4.00 ± 11.00			
	RUSF + HiML	25	3.00 ± 4.00			
Jumping and hopping (total/60) *^,$^	Baseline performance		Mean: 42.21	W(1) = 80.624, *p* < 0.001		
	No intervention	19	2.35 ± 1.34	W(2) = 44.110, *p* < 0.001	RUSF + HiML = RUSF RUSF + HiML > control RUSF > control	*p* = 0.421 *p* < 0.001 *p* < 0.001
	RUSF	21	12.26 ± 1.25			
	RUSF + HiML	23	13.68 ± 1.22			
	Sex * intervention			W(2) = 11.438, *p* = 0.010		

Abbreviations: RUSF: ready-to-use supplementary food, HiML: high-intensity motor learning, SD: standard deviation; IQR: interquartile range, * Wald statistics, ^$^ the baseline performance was added as a covariate.

## Data Availability

The corresponding author can provide the datasets upon reasonable request for data used during the analysis.

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
