# Peer review of "Effects of High-Intensity Motor Learning and Dietary Supplementation on Motor Skill-Related Physical Fitness in Thin Ethiopian Children Aged 5 to 7 Years: An Exploratory Pilot Cluster-Randomized Trial"

_nutrients, 2024, doi:10.3390/nu17010030_

Round 1

Reviewer 1 Report

Comments and Suggestions for Authors

The authors present a randomized controlled trial to evaluate the effects of high-intensity motor learning and dietary supplementation on motor skill-related physical fitness in 5-7 years old children with moderate thinness living in Southwest, Ethiopia. The topic is of great public health importance. I thank the authors for their efforts and contribution. Suggestions for improvement:

1) The authors randomized three schools into three arms (yielding one school per trial arm); they did not randomize the children. One of the major benefits of randomization is that if done properly, the randomization process balances confounders between trial arms... in this case, because the entire school was randomized to an arm, and there is only 1 school per arm, there could still be confounding due to imbalance in characteristics between the children in the 3 different schools. Therefore, this essentially becomes similar to a non-randomized study... indeed there are some differences between schools as seen in Table 1 which should be adjusted for in analyses, and then there is also the question about unmeasured confounding which is typically taken care of by proper randomization. If this were truly a cluster-randomized trial, then you would have multiple clusters per arm (not just one cluster per arm) and your power calculation methods would be different for a cluster-RCT. Some references say there should be a minimum of 4 clusters per arm in pilot RCTs... see following references for methodological recommendations for designing and conducting cluster-RCTs:

Hemming K, Taljaard M. Key considerations for designing, conducting and analysing a cluster randomized trial. Int J Epidemiol. 2023 Oct 5;52(5):1648-1658. doi: 10.1093/ije/dyad064. PMID: 37203433; PMCID: PMC10555937.

Hemming K, Taljaard M, Gkini E, Bishop J. Sample size determination for external pilot cluster randomised trials with binary feasibility outcomes: a tutorial. Pilot Feasibility Stud. 2023 Sep 19;9(1):163. doi: 10.1186/s40814-023-01384-1. PMID: 37726817; PMCID: PMC10507981.

2) Given that the school (and not the children) were randomized, the CONSORT flow diagram (figure 1) is inaccurate and misleading. Instead of saying Randomized n=75 children; the figure should say randomized 3 schools (containing total 75 children). Given there attrition after randomization from 75 to 69, please mention the number of children dropped out in each arm i.e. in each arm, give total n at the time of randomizing the school and then mention how many dropped out in each school.

3) Please provide a completed CONSORT checklist in the appendix.

4) Because there are three arms, please show power calculations for three arms (not two) and mention the minimum required sample size per arm. If underpowered, please mention that in the limitations section.

5) Please conduct regression analysis adjusting for covariates (because of incorrect randomization as explained in point 1)

6) In the limitations para, please mention the issues with randomization that are stated above in point 1, and mention the potential for unmeasured and residual confounding despite adjusting for covariates in the regression model in the revised analyses.

7) Please consider presenting this as a non-randomized study because of issues with randomization that are pointed out above in point #1

Thank you.

Author Response

December 12, 2024

Editor-in-Chief of the Journal of Nutrients

Subject: Submitting the revised version of our manuscript

Dear Editor,

We are submitting the revised version of our manuscript, "Effect of high-intensity motor learning and dietary supplementation on motor skill-related physical fitness in 5-7 year old children with moderate thinness living in Southwest, Ethiopia: a cluster-randomized controlled trial," Manuscript ID: nutrients-3360935, for publication in your journal.

We want to express our sincere gratitude to all the reviewers for their invaluable and constructive comments and feedback. We have considered each comment provided by the reviewers and addressed them comprehensively. We confirm that all co-authors have thoroughly reviewed this version of the manuscript. Please find a detailed point-by-point rebuttal to the reviewer's comments below. In the main document, we highlighted the changes in yellow. We believe that the revisions we made have substantially improved the manuscript and are hopeful that our manuscript is suitable for publication in your journal.

Best regards,

Melese Sinaga Teshome, on behalf of all co-authors

Independent Review Report, Reviewer: 1

Dear Reviewer, we appreciate all the invaluable comments that you gave on our manuscript. We considered all your comments and questions and carefully revised the manuscript. Below you can find a point-by-point response to your comments and questions. We believe that the revisions we made have substantially improved the manuscript.

Comments to the Author

Comments and Suggestions for Authors

The authors present a randomized controlled trial to evaluate the effects of high-intensity motor learning and dietary supplementation on motor skill-related physical fitness in 5-7 years old children with moderate thinness living in Southwest, Ethiopia. The topic is of great public health importance. I thank the authors for their efforts and contribution. Suggestions for improvement:

1) The authors randomized three schools into three arms (yielding one school per trial arm); they did not randomize the children. One of the major benefits of randomization is that if done properly, the randomization process balances confounders between trial arms... in this case, because the entire school was randomized to an arm, and there is only 1 school per arm, there could still be confounding due to imbalance in characteristics between the children in the 3 different schools. Therefore, this essentially becomes similar to a non-randomized study... indeed there are some differences between schools as seen in Table 1 which should be adjusted for in analyses, and then there is also the question about unmeasured confounding which is typically taken care of by proper randomization. If this were truly a cluster-randomized trial, then you would have multiple clusters per arm (not just one cluster per arm) and your power calculation methods would be different for a cluster-RCT. Some references say there should be a minimum of 4 clusters per arm in pilot RCTs... see following references for methodological recommendations for designing and conducting cluster-RCTs:

Hemming K, Taljaard M. Key considerations for designing, conducting and analysing a cluster randomized trial. Int J Epidemiol. 2023 Oct 5;52(5):1648-1658. doi: 10.1093/ije/dyad064. PMID: 37203433; PMCID: PMC10555937.

Hemming K, Taljaard M, Gkini E, Bishop J. Sample size determination for external pilot cluster randomised trials with binary feasibility outcomes: a tutorial. Pilot Feasibility Stud. 2023 Sep 19;9(1):163. doi: 10.1186/s40814-023-01384-1. PMID: 37726817; PMCID: PMC10507981.

Response: - We fully acknowledge the comment and agree to the concerns, which is why we mentioned this concern to the study limitations.

  • Due to resource limitations, we were not able to achieve the desired number of clusters (Schools) per arm indicating that the study might have power insufficiency. However, the schools were allocated to each arm randomly, and the study participants were selected randomly within each cluster. Based on the suggested literature we decided to adjust the model for baseline values of the outcome by adding the baseline results in the analysis of the covariance model as a covariate (Hemming 2023).
  • Further, we have acknowledged the limitations related to power in the discussion part of the manuscript. Given all the above limitations, the study is the first of its kind to show the effect of dietary supplements and HIML on children with moderate thinness. We have also recommended further study to redeem our limitations.

2) Given that the school (and not the children) were randomized, the CONSORT flow diagram (figure 1) is inaccurate and misleading. Instead of saying Randomized n=75 children; the figure should say randomized 3 schools (containing total 75 children). Given their attrition after randomization from 75 to 69, please mention the number of children dropped out in each arm i.e. in each arm, give total n at the time of randomizing the school and then mention how many dropped out in each school.

Response: - Thank you for this comment. We addressed this suggestion in the methods part of the manuscript.

  • Although 75 Children were identified to be involved in the study, six of them refused to participate. Therefore 69 children were randomized into the three arms as follows: RUSF (n=23), RUSF + HiML (n=25), and a control group (n=21). The text and the CONSORT Flow diagram are revised accordingly. There were no dropouts after randomization.

  • 3) Please provide a completed CONSORT checklist in the appendix.

Response: Thank you for this suggestion. We included the CONSORT checklist as an appendix in the manuscript.

4) Because there are three arms, please show power calculations for three arms (not two) and mention the minimum required sample size per arm. If underpowered, please mention that in the limitations section.

Response: - Thank you for this comment. The minimum sample size required per arm was mentioned in the methods part of the manuscript. The following paragraph is included in the discussion.

  • “This study has some limitations that must be acknowledged. One potential issue is social desirability bias, which was minimized by informing respondents that their answers were for comparison purposes only and would not affect service use or their privacy. Additionally, it is important to note that the children's dietary intake during the intervention was not monitored, and the sample size does not provide adequate statistical power for some of the PERF-FIT items. Contrarily, this may also indicate that these items do not reflect core difficulties in this particular patient group, especially, since many other items did reveal significant increases.”

5) Please conduct regression analysis adjusting for covariates (because of incorrect randomization as explained in point 1)

Response: - We chose to apply a derived form of regression analysis because we wanted to look into the differences between the groups instead of explaining the variance in the outcome. We therefore used AN(C)OVA.  

6) In the limitations part, please mention the issues with randomization that are stated above in point 1, and mention the potential for unmeasured and residual confounding despite adjusting for covariates in the regression model in the revised analyses.

Response: - As mentioned above (question 4), we included this limitation in the discussion section. Furthermore, as mentioned in point 5, we chose to apply AN(C)OVA to directly investigate differences between groups.  

7) Please consider presenting this as a non-randomized study because of issues with randomization that are pointed out above in point #1

Response: - We fully understand the reason for the comment. However, the schools are truly randomized into the three arms although the number of clusters per arm is not sufficient. We have explained this as a limitation related to power. Furthermore, we also used the term cluster RCT in our previously published paper on this trial, but with a different outcome. We therefore prefer to keep using the term cluster RCT, despite the fact that we agree with your comments.

Thanks so much 

Reviewer 2 Report

Comments and Suggestions for Authors

The purpose of the present study was to evaluate that bridge this gap by examining the combined effect of Ready-to-Use Supplementary Food (RUSF) and high-intensity motor learn-ing (HiML) on motor skill-related physical fitness in children with moderate thinness (MT). 

However, there are several revisions will required.

1) Authors mentioned that therefore, this study primarily investi- gated the effect of RUSF with(out) HiML compared to no intervention on motor skill- Nutrients 2024, 16, x FOR PEER REVIEW 4 of 17 related physical fitness in 5-7-year-old MT children. However, there was no hypothesis in the present study. 

2) There was no show that methods of the sample size in the present study.

3) Figures are not easy to understand, so could you show them more clearly?

4) Are there more study limitations here? 

Author Response

December 12, 2024

Editor-in-Chief of the Journal of Nutrients

Subject: Submitting the revised version of our manuscript

Dear Editor,

We are submitting the revised version of our manuscript, "Effect of high-intensity motor learning and dietary supplementation on motor skill-related physical fitness in 5-7 year old children with moderate thinness living in Southwest, Ethiopia: a cluster-randomized controlled trial," Manuscript ID: nutrients-3360935, for publication in your journal.

We want to express our sincere gratitude to all the reviewers for their invaluable and constructive comments and feedback. We have considered each comment provided by the reviewers and addressed them comprehensively. We confirm that all co-authors have thoroughly reviewed this version of the manuscript. Please find a detailed point-by-point rebuttal to the reviewer's comments below. In the main document, we highlighted the changes in yellow. We believe that the revisions we made have substantially improved the manuscript and are hopeful that our manuscript is suitable for publication in your journal.

Best regards,

Melese Sinaga Teshome, on behalf of all co-authors

Independent Review Report, Reviewer: 2

Dear Reviewer, We appreciate all the invaluable comments you gave on our manuscript. We considered all your comments and questions and carefully revised the manuscript. Below, you can find a point-by-point response to your comments and questions. We believe that the revisions we made have substantially improved the manuscript.

Comments to the Author

Comments and Suggestions for Authors

The purpose of the present study was to evaluate and bridge this gap by examining the combined effect of Ready-to-Use Supplementary Food (RUSF) and high-intensity motor learn-ing (HiML) on motor skill-related physical fitness in children with moderate thinness (MT). 

However, there are several revisions will required.

1) Authors mentioned that therefore, this study primarily investigated the effect of RUSF with(out) HiML compared to no intervention on motor skill- Nutrients 2024, 16, x FOR PEER REVIEW 4 of 17 related physical fitness in 5-7-year-old MT children. However, there was no hypothesis in the present study. 

Response: - Thank you for this comment. The hypothesis is included in the introduction of the manuscript as follows:

  • “MT children 5–7years of age who received dietary supplementation (RUSF) alone or combined with functional high-intensity motor training have better motor skills and, muscular fitness compared to those who receive neither.”

2) There was no show that methods of the sample size in the present study.

Response: - Thank you for this comment. The sample size determination process is explained in the methods part of the manuscript as follows:

  • “In the absence of prior studies to guide the sample size calculation, we assumed a medium effect size of 0.5, a design effect of 1.5 to account for clustering at the school level, a 10% loss to follow-up, and a significance level of 5%. Using G*Power 3.1.9.4, the required sample size for an ANOVA with three arms and 80% power was estimated at 42 participants. After adjusting for the design effect and expected loss to follow-up, the final sample size was calculated to be 69.”

3) Figures are not easy to understand, so could you show them more clearly?

Response: - Thank you for your suggestion and comment accepted. We modified the figures and included them in the manuscript.

  • Now we increased the size of figures 1 and 2.

4) Are there more study limitations here? 

Response: - Thank you for this comment. More limitations of the study are included in the discussion part of the manuscript based on the comments. The following paragraph is included in the discussion.

  • “This study has some limitations that must be acknowledged. One potential issue is social desirability bias, which was minimized by informing respondents that their answers were for comparison purposes only and would not affect service use or their privacy. Additionally, it is important to note that the children's dietary intake during the intervention was not monitored, and the sample size does not provide adequate statistical power.”

Thanks so much

Reviewer 3 Report

Comments and Suggestions for Authors

Title and Abstract

Title: The title is informative but overly detailed. Simplify to "Effects of High-Intensity Motor Learning and Dietary Supplementation on Motor Fitness in Ethiopian Children: A Cluster-Randomized Trial." This will enhance readability while maintaining clarity.

Abstract: The abstract effectively summarizes the study but lacks detail in methodology and significance. Add a brief mention of the statistical methods used (e.g., GEE) and clarify the primary and secondary outcomes.

Introduction

The introduction provides a robust background but could benefit from clearer connections between malnutrition, motor skills, and the study's novelty. Explicitly emphasize how this study fills existing gaps in the literature, particularly on combining nutritional and physical interventions.

Methods

Study Design: Clearly outlined, but specify why three arms were chosen and justify the lack of double-blinding.

Participants: Inclusion and exclusion criteria are thorough. Address potential biases due to the exclusion of children with medical conditions.

Sample Size: The calculation is well-documented but clarify if adjustments for potential cluster effects were adequately considered.

Intervention Description: Provide more detail on how HiML training was tailored to this population and whether adherence was consistently monitored.

Outcome Measures: The choice of PERF-FIT is appropriate, but discuss why it was selected over alternative measures and its sensitivity in malnourished populations.

Statistical Analysis: Explain why GEE was preferred over mixed models and address how missing data were handled.

Results

The results are well-organized with appropriate statistical analysis. However:

Include effect sizes for key findings to indicate clinical relevance.

Expand on subgroup analyses (e.g., sex and age) with more granular data in supplementary material.

Clarify the interpretation of non-significant findings, such as the lack of improvement in running and balance.

Discussion

Strengths: Highlighted effectively but could expand on how this study advances nutritional and rehabilitation practices in resource-limited settings.

Limitations: Acknowledge additional limitations, such as the short intervention duration and lack of long-term follow-up.

Comparative Analysis: The discussion compares findings to prior studies but could integrate more recent evidence on motor skill training in malnourished children.

Mechanisms: Expand on the physiological rationale linking RUSF and HiML to motor skill improvement, possibly citing more recent neuroscience or nutrition studies.

Recommendations

Suggest exploring long-term impacts and scalability in future studies. The paper briefly mentions school integration but lacks practical implementation strategies for low-resource settings.

Conclusion

Concisely written but consider adding a direct statement on policy implications for integrating nutritional and motor training programs in schools.

Figures and Tables

Figures are clear but ensure legends provide sufficient detail for standalone interpretation.

Tables effectively summarize findings but consider adding confidence intervals alongside p-values for precision.

References

Comprehensive but slightly dated. Incorporate more recent (post-2020) studies, especially systematic reviews or meta-analyses relevant to malnutrition and motor skill interventions.

Author Response

December 12, 2024

Editor-in-Chief of the Journal of Nutrients

Subject: Submitting the revised version of our manuscript

Dear Editor,

We are submitting the revised version of our manuscript, "Effect of high-intensity motor learning and dietary supplementation on motor skill-related physical fitness in 5-7 year old children with moderate thinness living in Southwest, Ethiopia: a cluster-randomized controlled trial," Manuscript ID: nutrients-3360935, for publication in your journal.

We want to express our sincere gratitude to all the reviewers for their invaluable and constructive comments and feedback. We have considered each comment provided by the reviewers and addressed them comprehensively. We confirm that all co-authors have thoroughly reviewed this version of the manuscript. Please find a detailed point-by-point rebuttal to the reviewer's comments below. In the main document, we highlighted the changes in yellow. We believe that the revisions we made have substantially improved the manuscript and are hopeful that our manuscript is suitable for publication in your journal.

Best regards,

Melese Sinaga Teshome, on behalf of all co-authors

Independent Review Report, Reviewer: 3

Dear Reviewer, we appreciate all the invaluable comments that you gave on our manuscript. We considered all your comments and questions and carefully revised the manuscript. Below you can find a point-by-point response to your comments and questions. We believe that the revisions we made have substantially improved the manuscript.

Comments to the Author

Comments and Suggestions for Authors

Title and Abstract

Title: The title is informative but overly detailed. Simplify to "Effects of High-Intensity Motor Learning and Dietary Supplementation on Motor Fitness in Ethiopian Children: A Cluster-Randomized Trial." This will enhance readability while maintaining clarity.

Response: - Thank you for the suggestion. We want to emphasize that we are including both motor skill competence and physical fitness and therefore find it important to keep the term ‘motor skill-related physical fitness’ as it was. We accepted the other suggestions to enhance the readability of the title:

  • "Effects of High-Intensity Motor Learning and Dietary Supplementation on Motor skill-related Physical Fitness in Thin Ethiopian Children aged 5 to 7 years: A Cluster-Randomized Trial."

Abstract: The abstract effectively summarizes the study but lacks detail in methodology and significance. Add a brief mention of the statistical methods used (e.g., GEE) and clarify the primary and secondary outcomes.

Response: - The comment was accepted and the following issues were included in the abstract part of the manuscript:

  • The primary outcome was motor skill-related physical fitness at baseline and endline using the Performance and Fitness test battery (PERF-FIT). The changes from baseline to end-line measurements were calculated as differences, and the mean difference in these changes/differences (DID) was then computed as the outcome measure. AN(C)OVA to directly investigate differences between groups. Statistical significance was declared at p-value ≤ 0.05

Introduction

The introduction provides a robust background but could benefit from clearer connections between malnutrition, motor skills, and the study's novelty. Explicitly emphasize how this study fills existing gaps in the literature, particularly on combining nutritional and physical interventions.

Response: - Thanks so much for your nice comments. Edits were made to the introduction to ensure connections between malnutrition, motor skills, and the study's novelty.

  • Given the combined effects of HTML and RUSF on children with moderate thinness and the lack of publications addressing how moderate thinness impacts gross motor development, muscular fitness, and motor skills—both globally and specifically in Ethiopia—it is essential to conduct a more in-depth analysis of this issue.

Methods

Study Design: Clearly outlined, but specify why three arms were chosen and justify the lack of double-blinding.

Response: - Thank you for this suggestion.

  • The study used three arms as there are two treatments and their combination and a control group. However, the fourth arm HIML only Could not be given to children with moderate thinness as it was suspected to aggravate the malnutrition unless it is given in combination with dietary supplementation. Therefore, the study continued with three arms. The study was single-blinded as it was not possible to mask it due to the supplements being given and the physical activity. However, the study participants are blind. That was why a cluster randomization was opted for.

Participants: Inclusion and exclusion criteria are thorough. Address potential biases due to the exclusion of children with medical conditions.

Response: - Thank you for this suggestion.

  • The reason for excluding children with medical conditions is that their treatment protocol differs from that of children with MT. Additionally, because the intervention includes HiML training, it may not be suitable for medically ill children.

Sample Size: The calculation is well-documented but clarify if adjustments for potential cluster effects were adequately considered.

Response: - Thank you for the comment. The sample size has considered a design effect of 1.5 to adjust for the potential effects of intra-cluster correlation.

Intervention Description: Provide more detail on how HiML training was tailored to this population and whether adherence was consistently monitored.

Response: - Thank you for this comment.

  • Before conducting the actual intervention of HiML, we carried out a feasibility study to assess the acceptability of the program among school children and to determine which types of play activities they prefer or dislike. Based on the findings from the feasibility study, we made modifications to the protocol to better align to the local context. During the 12-week intervention, all activities were monitored daily, and a regular discussion was held with trainers to evaluate whether the study participants were compliant. The attendance of the study participants was also documented in the study protocol.

Outcome Measures: The choice of PERF-FIT is appropriate, but discuss why it was selected over alternative measures and its sensitivity in malnourished populations.

Response: - Thank you for this suggestion.

  • We chose to use the PERF-FIT because this is the only available metric that allows assessment of both motor skill competence and muscular fitness. As such, it is more sensitive to capture the effect of malnutrition on gross motor performance and muscle function as wasting leads to loss of muscle mass.

Statistical Analysis: Explain why GEE is preferred over mixed models and address how missing data were handled.

Response: - Thank you for this critical question.

  • We do not have missing data. However, why we used generalized estimating equations is due to the interest in estimating population-level averages across the schools (the different arms) rather than individual-level random effects. GEE is better for this purpose.

Results

The results are well-organized with appropriate statistical analysis. However: Include effect sizes for key findings to indicate clinical relevance. Expand on subgroup analyses (e.g., sex and age) with more granular data in supplementary material. Clarify the interpretation of non-significant findings, such as the lack of improvement in running and balance.

Response: Thank you for the comment. The effect sizes are included in the results and clinical significance is included in the discussion.

  • We agree with the reviewer about the importance of zooming in on the subgroups more in detail. However, because of the small sample sizes, we chose not to do this as we may encounter power insufficiency. We have tried to see the outcomes by the age groups in both bivariate analyses and in the GEE model.

Discussion

Strengths: Highlighted effectively but could expand on how this study advances nutritional and rehabilitation practices in resource-limited settings.

Response: - Comment accepted and revised in the text of the manuscript.

  • The recent research on the same age group indicated that the combination of RUSF and HiML interventions improved the body composition, height, weight, and muscle strength of moderately thin children studied.

Limitations: Acknowledge additional limitations, such as the short intervention duration and lack of long-term follow-up.

Response: - Thank you for addressing the importance of the study limitation. In addition, the power issues that we included in the discussion section, we also mentioned the shortcoming of the lack of follow-up. However, we chose not to mention the short duration of the intervention, since 12 weeks is an acceptable time period for motor training as well as for RUSF, which should not be added for a longer period.

Comparative Analysis: The discussion compares findings to prior studies but could integrate more recent evidence on motor skill training in malnourished children.

Response: - Comment accepted, but there is no published paper on the subject to the best search literature.

Mechanisms: Expand on the physiological rationale linking RUSF and HiML to motor skill improvement, possibly citing more recent neuroscience or nutrition studies.

Response: -We acknowledge the comments and accepted them. included in the introduction part of the manuscript

  • Both RUSF and the HiML are expected to regenerate the wasted muscles of children with moderate thinness. While the RUSF supplies adequate nutrients including protein and other micronutrients, the HiML increases the muscle size due to excursion.

Recommendations

Suggest exploring long-term impacts and scalability in future studies. The paper briefly mentions school integration but lacks practical implementation strategies for low-resource settings.

Response: - Thank you for this suggestion. The practical implementation is included in the last part of the discussion as follows:  

  • This will help us gain insights into the reasons for low motor competence, the effects of inequalities on motor competence, and whether specific subgroups require targeted interventions to enhance the motor competence of children with MT.
  • It is also included in 4.2 Recommendations for future research and clinical practice

Conclusion

Concisely written but consider adding a direct statement on policy implications for integrating nutritional and motor training programs in schools.

Response: - Thanks so much for your comments and the comment is accepted and the following paragraph is included.

  • The findings of this study will assist policymakers, both governmental and non-governmental, in to Ethiopia, to implement appropriate policies to combat malnutrition effectively through national school programs.

Figures and Tables

Figures are clear but ensure legends provide sufficient detail for standalone interpretation.

Tables effectively summarize findings but consider adding confidence intervals alongside p-values for precision.

Response: - thanks so much for your comments and comments are accepted

  • In the supplementary material of Table A.3 was included

References

Comprehensive but slightly dated. Incorporate more recent (post-2020) studies, especially systematic reviews or meta-analyses relevant to malnutrition and motor skill interventions.

Response: - We acknowledged the comments and accepted them; however, we didn't recently find a published systematic review and meta-analysis related to our title but we got a single study that shows the impact of malnutrition on motor skills and the positive effect of play-based therapy.

  • This sentence is including in the discussion part of the manuscript ¨¨The study conducted in Ethiopia demonstrates that play-based psychomotor and psychosocial stimulation positively impacts the motor development (both fine and gross) of children with Severe Acute Malnutrition (SAM) aged 6 months to 6 years. This research indicates that play-based stimulation is beneficial for treating SAM children under six in low-income settings. The findings revealed a significant improvement in gross motor functions during the hospital stay, while fine motor functions showed improvement after discharge during home follow-up. Both younger and older children reaped similar benefits from the intervention (81). Additionally, a study conducted in South Africa found that moderate levels of stunting and wasting negatively affect the school performance and motor functioning of school beginners (82). A study conducted in South Africa indicated that moderate to vigorous physical activity levels were positively associated with health-related physical fitness (HRPF), motor-related physical fitness (MRPF), and certain motor skills in children aged 5 to 8 years (83)¨¨.

New add references

80.Hemming K, Taljaard M, Gkini E, Bishop J. Sample size determination for external pilot cluster randomised trials with binary feasibility outcomes: a tutorial. Pilot and Feasibility Studies. 2023 Sep 19;9(1):163.

81.Abessa TG, Worku BN, Wondafrash M, Girma T, Valy J, Lemmens J, Bruckers L, Kolsteren P, Granitzer M. Effect of play-based family-centered psychomotor/psychosocial stimulation on the development of severely acutely malnourished children under six in a low-income setting: a randomized controlled trial. BMC pediatrics. 2019 Dec;19:1-20.

82.Pienaar AE. Pienaar The association between under-nutrition, school performance and perceptual motor functioning in first-grade South African learners: The North-West Child Health Integrated with Learning and Development study. Health SA Gesondheid. 2019;24.

83.Gericke C, Pienaar AE, Gerber B, Monyeki MA. Relationships between moderate vigorous physical activity, motor-and health-related fitness and motor skills in children. African Journal of Primary Health Care & Family Medicine. 2024 May 20;16(1):4258.

Thanks so much

Round 2

Reviewer 1 Report

Comments and Suggestions for Authors

1) Abstract (line 34): "A total of 69 children were randomized into three intervention arms "  This sentence is incorrect and misleading because the children were not randomized but the school was. Please rephrase stating that the 3 schools were randomized to three arms.

2) Given that this study is clearly underpowered (even for a pilot cluster RCT for which the recommendation is a minimum of four clusters per arm- based on references shared before), please mention in the title and in the abstract that this is an 'exploratory pilot study'.

3) In your response you mentioned using AN(C)OVA, but in the statistical analysis section, the manuscript still mentions ANOVA ; please check and change it to ANCOVA. Please state (in the methods section as well as in the footnote under the table showing results) which additional variables were added as covariates in the ANCOVA model.

4) Please add in the limitations para that findings from this study must be considered as exploratory and hypothesis-generating. Please also add the following limitations: potential for (1) selection bias, (2) residual and unmeasured confounding, and (3) type I error owing to the sample size being just one cluster per arm.

Author Response

December 13, 2024

Editor-in-Chief of the Journal of Nutrients

Subject: Submitting the second revised version of our manuscript

Dear Editor,

We are submitting the revised version of our manuscript, "Effect of high-intensity motor learning and dietary supplementation on motor skill-related physical fitness in thin Ethiopian children aged 5 to 7 years: an exploratory  pilot cluster-randomized controlled trial," Manuscript ID: nutrients-3360935, for publication in your journal.

We want to express our sincere gratitude to the reviewer for their constructive comments and feedback. We confirm that all co-authors have thoroughly reviewed this version of the manuscript. Please find a detailed point-by-point rebuttal to the reviewer's comments below. In the main document, we highlighted the changes in yellow. We are hopeful that our manuscript is suitable for publication in your journal.

Best regards,

Melese Sinaga Teshome, on behalf of all co-authors 

Independent Review Report, Reviewer: 1

Dear Reviewer, we appreciate the comments that you gave on our manuscript. Below you can find a point-by-point response to your comments and questions. We believe that the revisions we made have substantially improved the manuscript. All changes that were made to the manuscript are highlighted in yellow.

Comments to the Author

1) Abstract (line 34): "A total of 69 children were randomized into three intervention arms "  This sentence is incorrect and misleading because the children were not randomized but the school was. Please rephrase stating that the 3 schools were randomized to three arms.

Response: Thank you for noticing this error. We added the correct wording here. This now reads ‘Three schools were randomized to three intervention arms, including a total of 69 children: RUSF (n=23), RUSF+HiML (n=25), and no intervention (n=21).’ And the flow diagram (Figure 1) was also updated to present the randomization process accordingly.

2) Given that this study is clearly underpowered (even for a pilot cluster RCT for which the recommendation is a minimum of four clusters per arm- based on references shared before), please mention in the title and in the abstract that this is an 'exploratory pilot study'.

Response: Thank you for this suggestion. We agree that, despite the large effect sizes, this type of design is underpowered as it is. We therefore added these terms to the title, as you suggested and also mentioned this in the discussion, where we discuss the strengths and limitations of our trial.

3) In your response you mentioned using AN(C)OVA, but in the statistical analysis section, the manuscript still mentions ANOVA ; please check and change it to ANCOVA. Please state (in the methods section as well as in the footnote under the table showing results) which additional variables were added as covariates in the ANCOVA model.

Response: Thank you for noticing this. We added the baseline result to the analysis as a covariate as suggested by the previously suggested literature, but only did so if there was a significant group effect present in the ANOVA. We therefore kept the description of the ANOVA in the statistical analysis section. We also referred to this paper in this section. In our previous version we mentioned the following: ‘In case of significant differences between the groups, the baseline results of the outcome were added as a covariate [80].” We now explicitly added that we conducted an ANCOVA. The text now reads: “In case of significant differences between the groups, an analysis of covariance (ANCOVA) was run where the baseline results of the outcome were added as a covariate [80].”

The results section was also updated with these new analyses as well as Table 2. We apologize for not doing that in the previous version. Additionally, we changed the figures (2 and 3), showing error bars per intervention arm, corrected for the mean of the significant covariates where applicable.

4) Please add in the limitations para that findings from this study must be considered as exploratory and hypothesis-generating. Please also add the following limitations: potential for (1) selection bias, (2) residual and unmeasured confounding, and (3) type I error owing to the sample size being just one cluster per arm.

Response: Thank you for pointing out this specific concern. We added the following sentences to the ‘strengths and limitations’ section: “Furthermore, due to the fact that we intended to avoid contamination between participants, we aimed to conduct a cluster RCT. However, since there is only one school per intervention arm, the current results should be interpreted with caution. The design as it is, despite its random selection of the children within a cluster, may have caused selection bias. The sample size does not provide adequate statistical power for this specific design increasing the risk of type I errors, and may have also caused residual and unmeasured confounding. Larger studies with multiple clusters per intervention arm are needed to verify the results reported in this exploratory pilot cluster RCT.”

Reviewer 3 Report

Comments and Suggestions for Authors

The revised manuscript, based on the highlighted changes and responses to reviewers' comments, demonstrates significant improvement. The authors addressed the reviewers' suggestions thoroughly, including title simplification, abstract detail enhancement, methodological clarifications, expanded discussion, and updated references. The manuscript now exhibits the following strengths:

Title and Abstract: The title is clear and retains necessary specificity. The abstract includes methodological details and primary outcomes, enhancing comprehension.

Introduction: Clearly emphasizes the study's novelty by linking malnutrition, motor skills, and physical fitness.

Methods: The study design and interventions are well-justified. Potential biases and limitations are acknowledged, with robust statistical methodology.

Results and Discussion: Results are well-documented, with effect sizes and significance detailed. The discussion integrates findings into existing literature and highlights both strengths and limitations effectively.

Figures, Tables, and References: Visuals and tables are clear, with added confidence intervals for precision. References were updated to include recent literature where applicable.

Author Response

Dear Reviwer, 

Thank you so much.